# A multivesicular body-like organelle mediates stimulus-regulated trafficking of olfactory ciliary transduction proteins

Devendra Kumar Maurya [1], Anna Berghard [1] & Staffan Bohm [1] ✉

Stimulus transduction in cilia of olfactory sensory neurons is mediated by odorant receptors, Gαolf, adenylate cyclase-3, cyclic nucleotide-gated and chloride ion channels. Mechanisms regulating trafficking and localization of these proteins in the dendrite are unknown. By lectin/immunofluorescence staining and in vivo correlative light-electron microscopy (CLEM), we identify a retinitis pigmentosa-2 (RP2), ESCRT-0 and synaptophysin-containing multivesicular organelle that is not part of generic recycling/degradative/exosome pathways. The organelle's intraluminal vesicles contain the olfactory transduction proteins except for Golf subunits Gγ13 and Gβ1. Instead, Gβ1 colocalizes with RP2 on the organelle's outer membrane. The organelle accumulates in response to stimulus deprivation, while odor stimuli or adenylate cyclase activation cause outer membrane disintegration, release of intraluminal vesicles, and RP2/Gβ1 translocation to the base of olfactory cilia. Together, these findings reveal the existence of a dendritic organelle that mediates both stimulus-regulated storage of olfactory ciliary transduction proteins and membrane-delimited sorting important for G protein heterotrimerization.

The morphology of neurons necessitates finely regulated endosomal transport systems that deliver selected cargos by long-distance anterograde/retrograde axonal and dendritic transport systems. Multivesicular bodies (MVBs) are important components of endosomal trafficking. MVBs are spherical organelles with a diameter of >250 nm that are characterized by a single outer ("limiting") membrane enclosing a variable number of cargo-containing intraluminal vesicles (ILVs)[1–3]. MVBs mediate protein sorting between early and late endocytic compartments and can return proteins to the plasma membrane or degrade their contents by fusing with lysosomes[4–7]. MVBs may also fuse with the plasma membrane and release their ILVs as exosomes. In addition, MVB-like organelles have specialized functions in certain cell types. For instance, MVB-like organelles are formed during melanosome biogenesis in melanocytes, and von Willebrand factor is stored in ILVs that are released as exosomes by platelets and endothelial cells[8,9]. Similarly, antigen presentation involves the class II MHC-containing late endosomes, which store and release class II MHCs at the plasma membrane[10]. Different subtypes of MVB-like organelles are also present in neurons, but their neuron-specific functions are largely unexplored[2,11–15]. However, a recent study by Ye et al. showed that a distinct type of MVB mediates retrograde NGF-TrkA signaling in sympathetic neurons[16].

Olfactory sensory neurons (OSNs) in the olfactory epithelium (OE) are dependent on specialized dendritic transport of signaling proteins. OSNs have a single dendrite that ends in a protrusion, the dendritic knob, which can be located 50–100 μm away from the cell body. Several thin cilia, emanate from the dendritic knob and are the sites of stimulus transduction initiation. The key ciliary proteins involved in olfactory transduction are odorant receptors (ORs), heterotrimeric olfactory Golf protein subunits (Gαolf, Gβ1, Gγ13), adenylate cyclase type 3 (AC3), cyclic nucleotide-gated ion channel subunit 2 (CNGA2) and the $Ca^{2+}$-activated chloride channel TMEM16B[17–22]. Olfactory cilia and the outer segments of photoreceptors mediate stimulus transduction by similar mechanisms, as e.g. ORs belong to the "rhodopsin-like" class of G protein-coupled receptors (GPCRs) and Gβ1 is a subunit of both transducin and Golf[17,18]. Moreover, visual and olfactory CNG

[1]Department of Molecular Biology, Umeå University, Umeå SE901 87, Sweden. ✉e-mail: staffan.bohm@umu.se

channels are composed of homologous subunits[23]. Retinitis pigmentosa-2 (RP2) is implicated in localizing opsin to the outer segment, and mutation of this protein causes the eye disease X-linked retinitis pigmentosa[24,25]. RP2 is a GTPase-activating protein for ADP ribosylation factor-like GTPase 3 (ARL3), which in turn stimulates the release of lipidated membrane proteins from carrier proteins such as the prenyl-binding protein phosphodiesterase subunit delta (PDEδ)[26–28]. In vitro analyses have shown that ARL3 also releases Gβ1 bound to RP2[29].

Studies of intracellular OR trafficking have revealed that proteins such as REEP and RTP1/2 promote the plasma membrane localization of ORs in heterologous cell types by regulating their exit from the ER or Golgi[30,31]. Other studies have identified proteins and mechanisms for ciliary targeting and intraflagellar transport of G protein subunits, CNGA2 and AC3, as well as for desensitization of transduction by β-arrestin2- and clathrin-mediated internalization of ORs into endosomes[18,32–36]. However, the pathways responsible for transduction protein storage and transport from the ER/Golgi in the soma to sites of ciliary targeting in the dendritic knob are unknown.

Here, we analyze OSNs in vivo by double/triple immunohistochemistry/lectin staining and correlative light–electron microscopy (CLEM), which combines information about protein localization obtained by confocal fluorescence microscopy with high-resolution cellular ultrastructure data obtained by electron microscopy[37]. We identify a dendritic MVB-like organelle that we have named the multivesicular transducosome (MVT). MVTs carry ORs, Gαolf, Gβ1, AC3, CNGA2 and TMEM16B and constitutively express ESCRT-0, synaptophysin and RP2. RP2 is associated with Gβ1 in the MVT´s limiting membrane. Odor stimuli as well as direct stimulation of the transduction pathway with forskolin result in disintegration of the limiting membrane, release of ILVs into the dendroplasm and translocation of RP2/Gβ1 to the plasma membrane and dendritic knob. Taken together, these results reveal a previously unrecognized molecular system that is distinct from prototypical MVB pathways and that, in a stimulus-dependent manner, controls the transport, storage and sorting of proteins that mediate GPCR signaling in neurons.

## Results

### Several ciliary olfactory transduction proteins colocalize in putative dendritic vesicles

We previously showed that AC3 and OR can colocalize in dendrosomatic puncta in OSNs[38]. To investigate whether these putative vesicles contain additional olfactory transduction proteins, we performed double and triple immunohistochemistry with antibodies that recognize AC3, ORs, CNGA2, Gαolf, Gβ1 and TMEM16B. The anti-OR antibody we used recognizes two closely related ORs (M71 and M72[39]). The results showed that all six transduction proteins colocalized in a few large distinct puncta in dendrites (arrows in Fig. 1b–e). Unlike the other transduction proteins, Gβ1 was also present in the plasma membrane of apical dendritic segments (arrowhead Fig. 1d). In the soma and the part of the dendrite close to the dendritic origin, smaller puncta that did not contain all transduction proteins were present (arrowheads Fig. 1b, c). Unlike canonical olfactory transduction in cilia, transient receptor potential cation channel C2 (TRPC2)-dependent transduction in response to $H_2S$ occurs in the dendritic knob[40]. Notably, double immunohistochemistry for TRPC2/CNGA2 and TRPC2/AC3 showed granular TRPC2 signals that did not colocalize with CNGA2 or AC3 (Supplementary Fig. 1). These results indicated that proteins that mediate stimulus transduction within the ciliary compartment, but not in the dendritic knob, were colocalized in putative large dendritic vesicles.

### Identification of an AC3[+] MVB-like dendritic organelle by CLEM

To investigate the ultrastructure of the multi-transduction protein-containing putative vesicles, we performed CLEM of mouse OE

sections. The most commonly used markers for CLEM are recombinant proteins with fluorescence tags that can be detected by transmission electron microscopy (TEM) via immunogold or quantum dot labeling[37]. Because genetically modified mice expressing fluorescence-tagged olfactory transduction proteins are not available, we developed a CLEM protocol using conventional immunohistochemistry to visualize OSNs in vivo, an approach that also avoids artifacts due to in vitro culturing, protein modification or overexpression. We successfully developed a protocol through which we could detect AC3 in ultrathin tissue sections by immunofluorescence. The results showed that AC3 was confined to large spherical MVB-like organelles characterized by a single limiting membrane enclosing small spherical ILVs (Fig. 1f–h´). Approximately half of the MVB-like organelles were densely packed with ILVs, while the other half had a clear central region without ILVs (Fig. 1f–h, f´–h´, Supplementary Figs. 2d, e and 3a–b´´). The mean diameter of these organelles was $662 \pm 16$ nm (Fig. 2a, b), which was large relative to the previously reported MVB diameters of 256 nm and 400–500 nm in cortical neurons and other cell types, respectively[15,41–43]. The mean diameter of the ILVs inside the organelles was $66 \pm 0.5$ nm, which is similar to the previously reported size of 50–80 nm[11] (Fig. 2b). The thicknesses of the limiting and ILV membranes were $6.7 \pm 0.1$ and $7.05 \pm 0.06$ nm, respectively (Fig. 2b), and the average number of ILVs per MVB-like organelle was $36 \pm 12$ per section (Fig. 2b). The MVB-like organelles lacked luminal tubules or tubular protrusions, which are associated with sorting cargo, protein recycling and transport toward the plasma membrane or the trans-Golgi network[2,15,44,45]. CLEM and TEM also showed that the soma contained lysosomes and autophagosome-like ultrastructures (Supplementary Fig. 3c). These organelles were restricted to the soma, which was in line with the finding that the lysosomal-associated membrane proteins LAMP1 and LAMP2 were also restricted to the soma, as indicated by immunofluorescence (Supplementary Fig. 4). Immunofluorescence also revealed that the transduction proteins in the soma both did and did not colocalize with LAMP1/2 (Supplementary Fig. 4). These results indicated that the MVB-like organelles in dendrites were not associated with a vesicular degradation pathway.

### Lectin staining and CLEM confirm that the large distinctive transduction protein-positive puncta are MVTs

The CLEM protocol we used to correlate AC3 immunofluorescence with the MVB-like organelles was not compatible with a number of different antibodies that recognize other olfactory transduction proteins. However, since AC3, M71/72, CNGA2, TMEM16B, Gαolf and Gβ1 were colocalized in dendrites, it is highly likely that the dendritic AC3[+] MVB-like organelles identified by CLEM also contained multiple transduction proteins. To confirm this assumption, we performed CLEM using two different biotinylated lectins: *Dolichos biflorus agglutinin* (DBA) and *Wisteria floribunda* (WFA). By using biotinylated lectins and fluorescence-tagged streptavidin, it is possible to visualize carbohydrate structures in glycoproteins with high efficiency. The rationale for using these lectins is that both are known to exhibit the same distribution as ORs in OE and OR-specific axonal projections to target neurons in the olfactory bulb of a small number (~5–10) of OR-specific OSNs, which are distinct from M71/72-expressing OSNs[46,47]. Thus, these lectins can be used as markers of ORs. The similarity between DBA staining and OR staining was further verified by our finding that DBA immunofluorescence in cilia was reduced in response to inhibition of hedgehog signaling in the same way we previously observed for OR immunofluorescence[38] (Supplementary Fig. 5). Collectively, these results indicated that DBA and WFA recognize carbohydrates that are associated with a select group of ORs, which is in line with the fact that N-terminal glycosylation has been shown to be critical for membrane trafficking of ORs[48]. Importantly, similar to ORs, we found that both DBA and WFA colocalized with AC3 and CNGA in dendrites and cilia

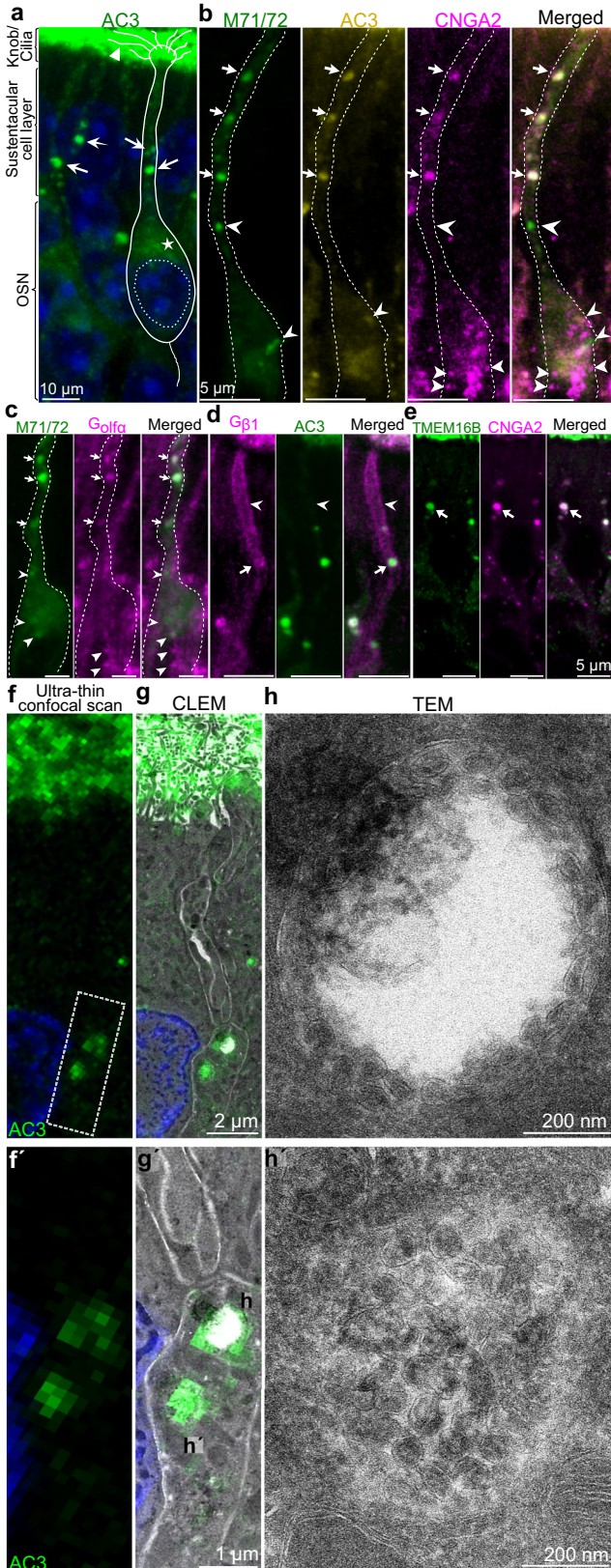

**Fig. 1 | Identification of MVB-like organelles containing several ciliary olfactory transduction proteins. a** Representative image showing AC3 immunofluorescence in a coronal OE section imaged by confocal microscopy. The OE cell layers are indicated in the margin. The soma, nucleus, axon, dendrite, knob and cilia of one OSN are outlined. AC3 immunofluorescence is present in cilia (arrowhead), the soma (asterisk) and dendritic puncta (arrows). **b, c** Images of triple (2 mice; 18 OSNs) and double immunohistochemistry (3 mice; 30 OSNs) showing colocalization of M71/72, AC3 and CNGA2 (**b**) as well as M71/72 and Gαolf (**c**) in three dendritic puncta (arrows). In some puncta located in the soma and close to the dendritic origin, the fluorescence did not overlap (arrowheads). **d** Results of double immunohistochemistry (2 mice; >30 OSNs) showing colocalization of Gβ1 and AC3 in dendritic puncta (arrow). Gβ1 but not AC3 was also present in the plasma membrane of the distal dendritic segment (arrowhead). **e** Images of double immunohistochemistry (2 mice; >25 OSNs) showing colocalization of CNGA2 and TMEM16B in dendritic puncta (arrow). **f–h** Representative images (4 mice; 17 OSNs) showing AC3 immunofluorescence (**f**) and CLEM images (**g**) of ultrathin (75 nm) sections through an OSN dendrite. The outlined area in (**f**), which contains two immunofluorescent puncta, is enlarged in f´ and g´. **h** TEM images showing that the two AC3-positive puncta (**h**, **h´**) corresponded to MVB-like organelles. One MVB-like organelle lacked ILVs in the center (**h**), whereas the other was densely packed with ILVs (**h´**). An enlarge overview of (**g**) is shown in supplementary fig. 2d.

identified by conventional confocal microscopy were MVB-like organelles. Given that these organelles carried several different transduction proteins, we named them MVTs.

### Each M71/72-expressing OSN contains a few MVTs in the proximal half of the dendrite

Because each OSN expressing the OR M71 or M72 is surrounded by OSNs expressing other ORs, we were able to examine the number and distribution of MVTs in individual OSNs. The results showed that the majority of M71/72-positive OSNs (> 90%) contained one or up to three distinctive MVTs that were restricted to a defined region located approximately in the middle of the dendrite, i.e., at an average position located approximately 40% of the dendrite length distal to the soma (Fig. 3a, b). The majority of OSNs also contained a variable number of smaller and less intense M71/72+ puncta both above and below the MVTs (Fig. 3a, c, d).

### Identification of RP2 in the limiting membrane of MVTs

To characterize the MVTs in dendrites in more detail, we performed systematic immunohistochemical analyses with antibodies that recognize proteins (i) that are known to label MVBs, (ii) that regulate vesicle transport and iii) that are implicated in trafficking of glycoproteins or ORs. The following antibodies were used; CD9, PDEδ, β-arrestin2, RAB7, RTP, AP1G1, Clathrin, LMAN2L, COPII (SEC23), SEC31, GRASP55, GRASP65, RAB10 and RAB35, IFT20 and IFT88. All antibodies labeled organelles or cellular structures unrelated to MVTs (Supplementary Fig. 1). We previously showed that dendritic ORs do not colocalize with RAB4A, RAB6B, RAB8A/B, RAB11A/B or the Golgi markers GM130 and TGN46[38]. Together these results strengthened our notion that the MVT was distinct from prototypic MVBs that form parts of lysosomal/recycling/exocytic pathways.

Immunohistochemical analysis of the MVTs identified RP2 as an MVT-associated protein. Immunofluorescence analyses revealed a circular pattern of RP2 immunofluorescence enclosing the MVTs (arrows in Fig. 4a–c, g, h). This pattern of RP2 immunofluorescence was not observed in the distal dendritic segment (white arrowheads in Fig. 4a, c). Interestingly, the disappearance of RP2 membrane localization correlated spatially with the appearance of RP2 fluorescence in the plasma membrane, with the RP2 fluorescence gradually increasing in intensity toward the dendritic knob (Fig. 4a–e). This change in the subcellular localization of RP2 could be a consequence of RP2 translocation from the limiting membrane to the plasma membrane, in conjunction with disintegration of the limiting membrane of the MVT.

(Fig. 2c–g). Moreover, the results of CLEM showed that the labeling efficiencies of both DBA and WFA were high even after tissue preparation for CLEM and that both lectins colocalized specifically with the MVB-like organelles in dendrites (Fig. 2c–g). Thus, although the exact targets of DBA and WFA are unknown, the results verified that the transduction protein-containing putative vesicles that we

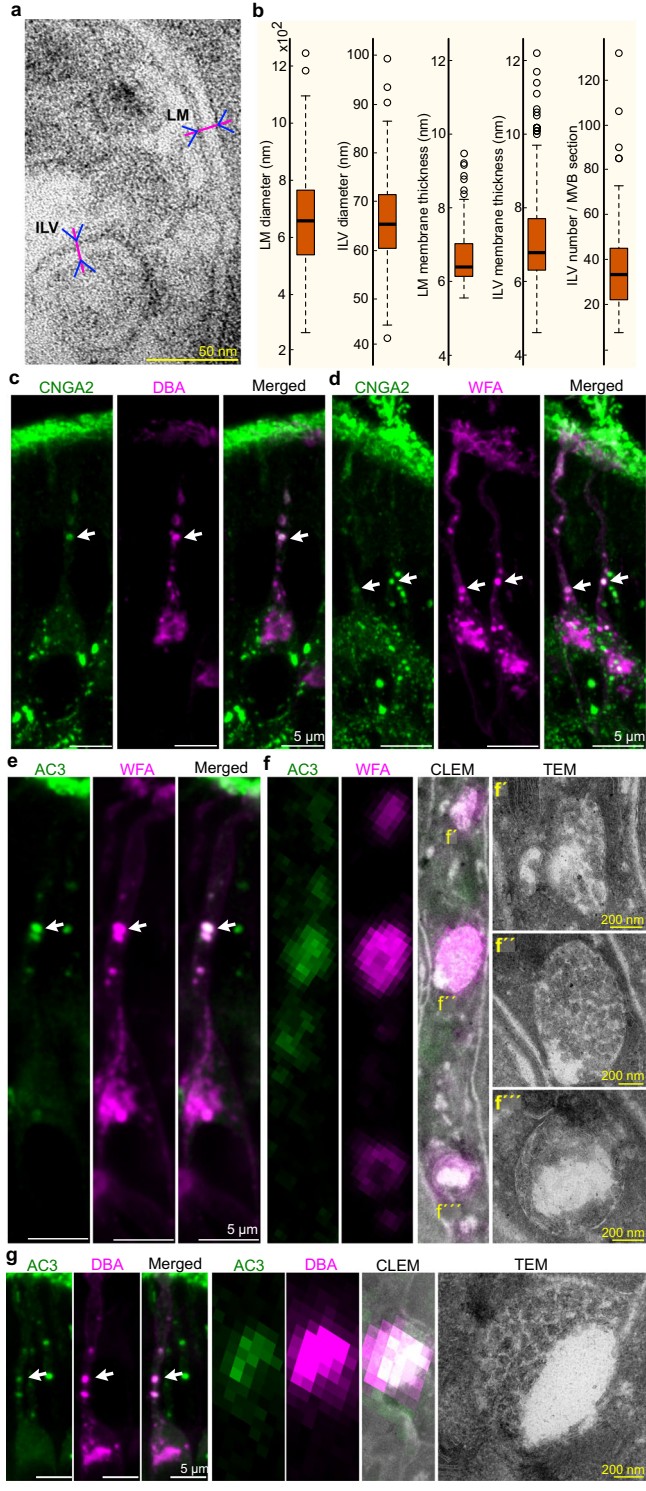

**Fig. 2 | Characterization of MVTs and analysis of lectin staining by CLEM.**
**a** High-resolution TEM images showing the limiting membrane (LM) and ILV membranes with magenta and blue distance markers crossing the membranes in ultrathin section. **b** Boxplots showing the diameter, limiting membrane thickness and ILV membrane thickness as well as the number of ILVs per section of MVB-like organelles. Data is from 97 OSNs ($N = 7$ mice). Sample sizes were $n = 137$ (LM diameter), $n = 144$ (ILV diameter), $n = 66$ (LM membrane thickness), $n = 443$ (ILV membrane thickness) and $n = 135$ (ILV number per MVB section). The box in plots denotes the interquartile range (IQR) i.e. median (line within box) ± 25–75 percentiles. Whiskers show rest of the data distribution ($\pm 1.5 \times$ IQR) and outliers are indicated by circles. Data is in source data file. Density of ILVs per MVB-like organelle section, falling within the IQR, is $112 \pm 6$ ILVs/$\mu$m$^2$ ($n = 70$). Estimates of ILVs from four individual MVB-like organelles within this IQR, by analysis of 2-3 serial sections (representing about a third of the total volume), yielded the following numbers: 223, 175, 161 and 152 ILVs/MVB-like organelle. **c, d** Representative images of CNGA2 double immunohistochemical staining with DBA (**c**; 2 mice, 8 OSNs) and WFA (**d**; 2 mice, 16 OSNs) are shown. Both lectins colocalized with CNGA2 in dendrites (arrows). **e** AC3 immunohistochemical staining/WFA staining analysis showing that AC3 and WFA were colocalized in dendrites (arrow). **f** AC3$^+$ and WFA$^+$ dendritic vesicles, a CLEM image and a TEM image showing close ups of three vesicles (f´, f´´, f´´´) with MVB-like morphology are shown. **e, f** Representative images for 5 mice; >27 OSNs are shown. **g** AC3 immunohistochemical staining/DBA staining analysis showing that AC3 and DBA were colocalized in dendrites (arrow). A representative CLEM image and a TEM image showing a close up of an MVB-like body organelle with AC3 and DBA fluorescence (3 mice, 13 OSNs).

## ESCRT-0, but not other ESCRT complexes, localizes to the limiting membrane of MVTs

Sorting of cargo and ILVs during the formation of prototypical MVBs is regulated by ESCRT complexes[49]. Interestingly, we found that MVTs were positive for HGS and STAM1, which form the ESCRT-0 complex that initiates the sequential assembly of MVBs (Fig. 5a, b). The ILV* clusters released by MVTs in the distal dendritic segment did not contain ESCRT-0 proteins (Fig. 5c, d). Thus, ESCRT-0 localized together with RP2 in the limiting membrane of MVTs. However, while RP2 may have relocated upon membrane disintegration, ESCRT-0 was not enriched in the distal dendritic plasma membrane or dendritic knob (Fig. 5c, d). Further analysis showed that proteins of the ESCRT machinery downstream of ESCRT-0, such as ESCRT-I (TSG101) and ESCRT-III (CHMP1A and CHMP4B), were not localized to MVTs (white arrows in Fig. 5e–h). Moreover, ESCRT-IV (VPS4A) was not localized to MVTs but was highly enriched in the dendritic knob (asterisk in Fig. 5h).

## Evidence for the initiation of MVT biogenesis in the soma

CLEM provided evidence that biogenesis of MVTs, such as MVTs and clusters of putative future ILVs that were not enclosed by a limiting membrane and were juxtaposed to the Golgi/ER, occurred in the soma (Supplementary Fig. 6c, d). Consistent with the presence of nascent MVTs in the soma, we observed transduction proteins both colocalized and not colocalized with RP2 (Fig. 4g, h). Conversely, we also identified a ring-shaped RP2 immunofluorescence signal without a transduction protein-positive core (green arrowhead in Fig. 4g). We speculated that these RP2-positive vesicles were nascent MVTs that did not contain all types of transduction proteins and therefore were only positive for RP2. This conclusion was supported by results of triple and double immuno/lectin staining as well as CLEM analyses, which indicated that vesicles not containing all transduction proteins were present in the soma and the dendritic origin (Figs. 1b, c, 2c, d and f´´´). As in dendrites, ESCRT-0 colocalized with M71/72 in the soma (Fig. 5a, b). However, unlike in dendrites, small M71/72-containing puncta that colocalized with ESCRT-III were also found in the soma (yellow arrows in Fig. 5f, g). In addition, the soma contained small M71/72$^+$ puncta that did not colocalize with ESCRT-0, ESCRT-III or ESCRT-IV and small ESCRT-0$^+$, ESCRT-III$^+$ and ESCRT-IV$^+$ puncta that did not colocalize with

In agreement with this hypothesis, CLEM showed that distal dendrites contained ILV clusters devoid of a limiting membrane (Fig. 4f, Supplementary Fig. 6a, b). By definition, loss of the limiting membrane meant that these structures could not be called MVTs. Furthermore, we could not call the released cargo-containing vesicles ILVs. For clarity, we use the term "ILV*" for small transduction protein-containing vesicles located in the dendroplasm. The ILVs* could appear in clusters that spread out from the electron-lucent luminal space of former MVTs (Supplementary Fig. 6a). Taken together, these results indicated that MVTs "burst" and released ILVs at approximately the middle of dendrites.

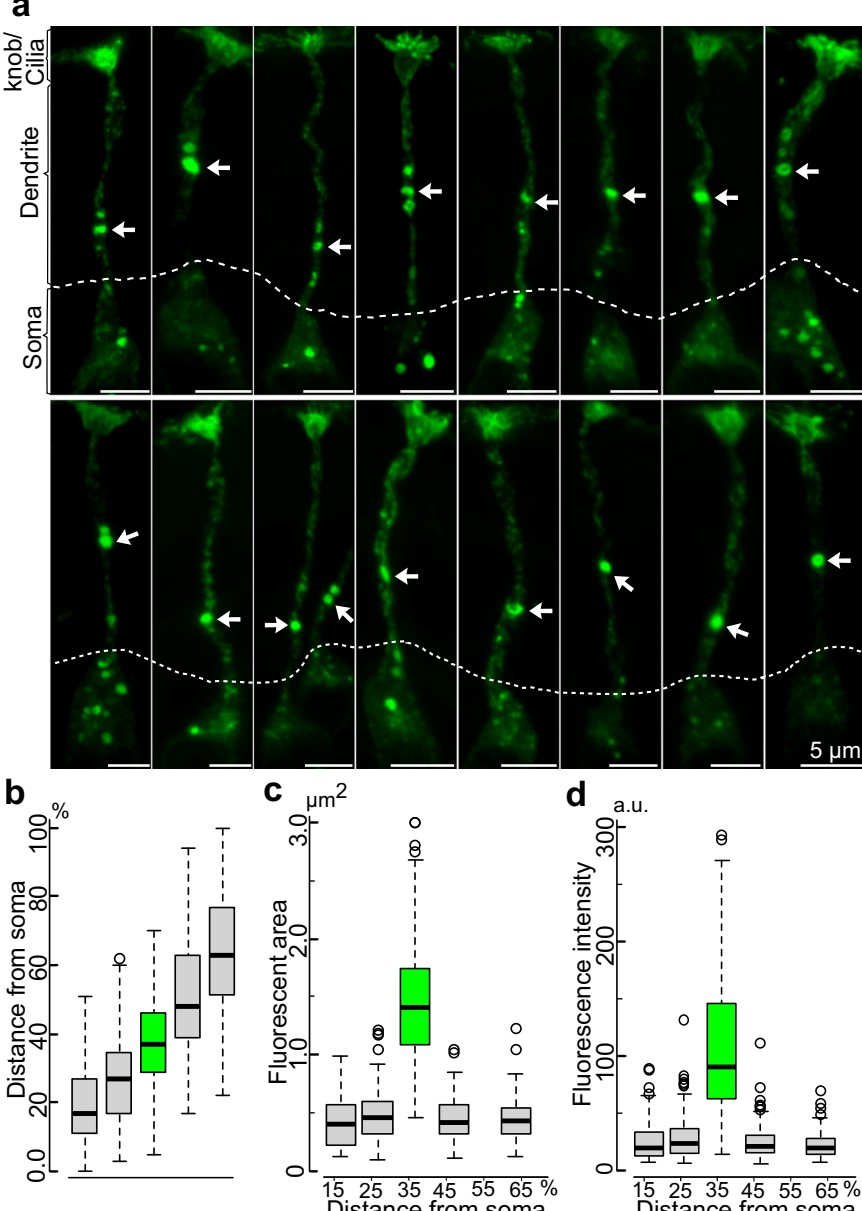

**Fig. 3 | Each OSN contains a few MVTs in a defined dendritic region.**
**a** Immunohistochemical analysis of ORs in individual OSNs are shown. Each OSN usually harbored 1-3 MVTs (arrows) localized to the proximal part of the dendrite. Above and below these puncta were many small and faintly stained M71/72$^+$ puncta. The dotted line indicates the dendritic origin. **b** Boxplot showing the distance from M71/72$^+$ puncta to the dendritic origin normalized to the percentage of the total length of the dendrite. The mean distance of MVTs distal to the dendritic origin shown in (**a**) (arrows) was 37% of the dendritic length. **c**, **d** Boxplot showing the fluorescent area (**c**) and intensity (**d**) of M71/72$^+$ puncta at different distances from the dendritic origin. **b**–**d** Boxplots are representing $N = 5$ mice; n = 166 OSNs. The box in plots denotes the IQR i.e. median (line within box) ± 25–75 percentiles. Whiskers show rest of the data distribution (±1.5 × IQR) and outliers are indicated by circles. The mean fluorescent area and intensity of the MVTs shown in (**a**) (arrows) were 1.41 μm$^2$ and 87.32 (arbitrary units, a.u.), respectively.

M71/72 (green and red arrowheads in Fig. 5f–h). These results indicated that ESCRT-dependent biogenesis of MVBs, possibly including also MVTs, occurred in the soma. Taken together, these results were in line with the hypothesis that vesicles containing a limited number of different olfactory transduction were generated in the soma and that subsequent homotypic fusion of these vesicles gave rise to the MVT. The extent to which ESCRT proteins regulate this process remains to be determined.

**Localization of RP2-binding proteins and synaptophysin**
To obtain evidence for a possible causative relationship between known RP2 functions and MVT assembly and disintegration, we

analyzed the expression of known RP2 binding partners. RP2 binds to ARL3 between the inner and outer segments of photoreceptors and regulates the unloading of cargo proteins such as opsin and G proteins to the outer segment[26–28,50]. We found that in OSNs, ARL3 was not localized to and was not in close proximity to MVTs (Supplementary Fig. 7a). Instead, RP2 and ARL3 immunofluorescence overlapped at the ciliary base in the dendritic knob, where the ARL3 effector PDEδ was also found (Supplementary Fig. 7a, c).

Another RP2-binding protein in the inner segment of photoreceptors is NSF, which is an AAA ATPase that plays a role in the regulation of synaptic vesicle fusion with the presynaptic plasma membrane[51,52]. NSF immunofluorescence showed a dotted pattern

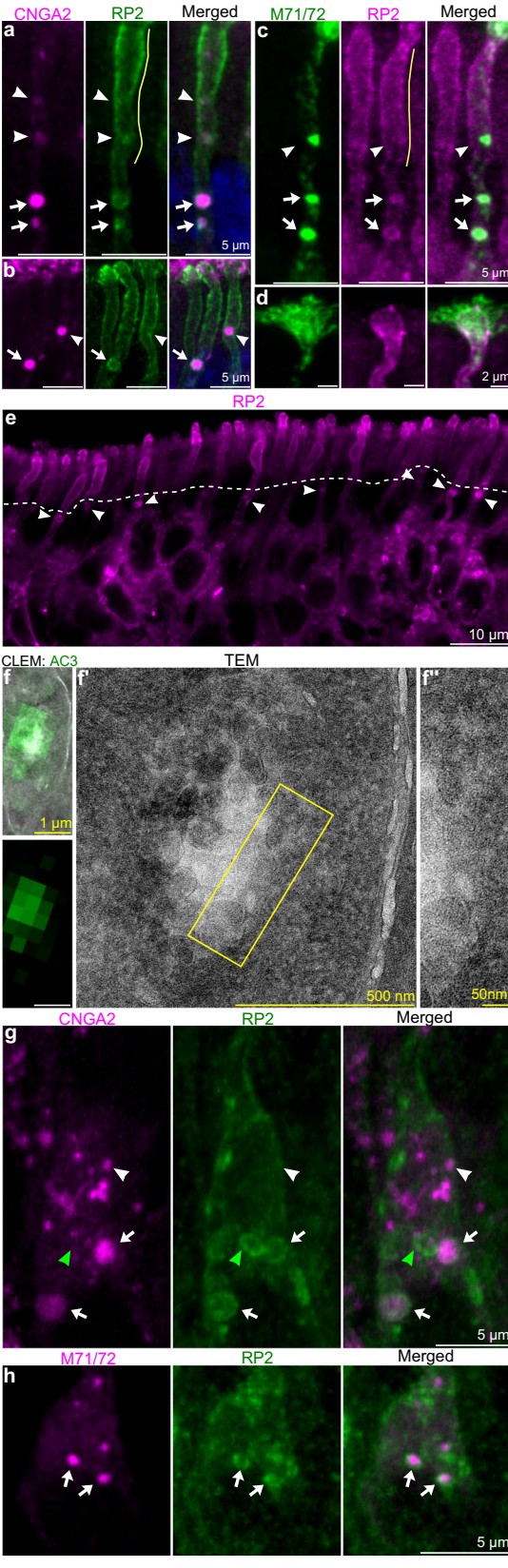

**Fig. 4 | The MVT is associated with RP2. a–c** Representative double immunohistochemistry images (5 mice) showing colocalization of RP2/CNGA2 (**a**, **b**) and RP2/M71/72 (**c**; 4 mice, >50 OSNs) in the limiting membrane of MVTs (arrows). CNGA2 and M71/72 fluorescent puncta partly overlapping with or lacking RP2 fluorescence are also shown (arrowheads). These puncta were located in the distal dendritic segment, in which RP2 fluorescence in the plasma membrane gradually increased (yellow lines in **a**, **c**) toward the dendritic knob (**d**). **e** Representative (5 mice, >20 scans) immunohistochemical image showing RP2 fluorescence in MVTs (arrowheads), the plasma membrane and the dendritic knob. The dashed line shows the border between intense and less intense RP2 fluorescence in the distal and proximal dendritic segments, respectively. **f**, Results of CLEM analysis (4 mice, >20 OSNs) of a dendrite showing a cluster of ILVs* (i.e., AC3⁺ vesicles without a limiting membrane). A close up of area outlined in f´ is shown in f´´. **g**, **h** Double immunohistochemistry showing colocalization of CNGA2/RP2 (**g**) and M71/72/RP2 (**h**) in the soma (arrows). The white and green arrowheads indicate CNGA2⁺/RP2⁻ and CNGA2⁻/RP2⁺ puncta, respectively. Shown are representative images (5 mice, >50 OSNs).

Interestingly, synaptophysin was also localized to MVTs (Supplementary Fig. 7e).

## Evidence that Golf subunits are not located together in dendrites

Schwarz et al. showed that RP2 binds specifically to Gβ1 in photoreceptors and that this interaction most likely facilitates membrane association and trafficking of Gβ1 prior to Gβ1:Gγ1 heterodimer formation[50]. Double immunohistochemical staining for RP2 and Gβ1 and line scans of immunofluorescence intensity showed that RP2 and Gβ1 fluorescence overlapped precisely in the limiting membrane as well as along the plasma membrane (Fig. 6a, b). RP2 and Gβ1 were also colocalized at the ciliary base in the dendritic knob (asterisk Fig. 6a). This result indicated that RP2 might regulate membrane association and trafficking of Gβ1 not only in photoreceptors but also in OSNs. Since RP2 has been shown to regulate the assembly of the Gβ1:Gγ1 heterodimer in photoreceptors, we analyzed Gγ13 expression[50]. Interestingly, Gγ13 was not localized in MVTs or in the dendritic plasma membrane (arrows in Fig. 6c). Instead, Gγ13 immunofluorescence was observed in the dendritic knob and cilia as well as in dendrosomatic puncta that were unrelated to MVTs.

## Release of ILVs from MVTs is olfactory stimulus dependent

The results indicated that ILVs were released from MVTs in the middle of dendrites as a consequence of disintegration of the limiting membrane. To address whether the release of ILVs is regulated by activation of OSNs by odor stimuli, we analyzed unilateral naris occluded mice[54]. Naris occlusion results in sensory deprivation of OSNs ipsilateral to the occluded naris, while OSNs on the non-occluded side receive twice the volume of inhaled air with odors. S100A5 immunohistochemistry was used to confirm successful naris occlusion, as S100A5 is transiently expressed in response to odor stimuli[55] (Fig. 7a). First, the number of MVTs (i.e., double RP2⁺/CNGA2⁺ puncta) in the OE of both nasal cavities was quantified in 12 days old mice that had been subjected to unilateral naris occlusion from postnatal day 5 (Fig. 7b, c). Sensory deprivation increased both the number of OSNs with MVTs and the number of OSNs with >3 MVTs (Fig. 7b, c). These results indicated that stimulus deprivation led to an accumulation of MVTs. We next quantified the ratio of inactive (S100A5⁻) OSNs with MVTs (red arrow in Fig. 7d´) to active (S100A5⁺) OSNs without MVTs (green arrows in Fig. 7d´) in OE on the occluded and non-occluded side. The results showed that naris occlusion increased the percentage of inactive OSNs with MVTs and decreased the percentage of active OSNs without MVTs (compare the blue and green sectors of the pie charts in Fig. 7e). The naris occlusion effect was evident not only relative to the non-occluded side, but also to OE in control (unoperated) mice. This indicated that doubling of the odor/air stream on the non-occluded

in the dendrites (Supplementary Fig. 7d). Even though NSF immunofluorescence was not restricted to MVTs, there was partial overlap, suggesting that NSF could potentially interact with RP2 in the limiting membrane. Synaptophysin is a protein that, like NSF, participates in the fusion of internal membranes by regulating soluble NSF attachment receptor (SNARE) proteins[53].

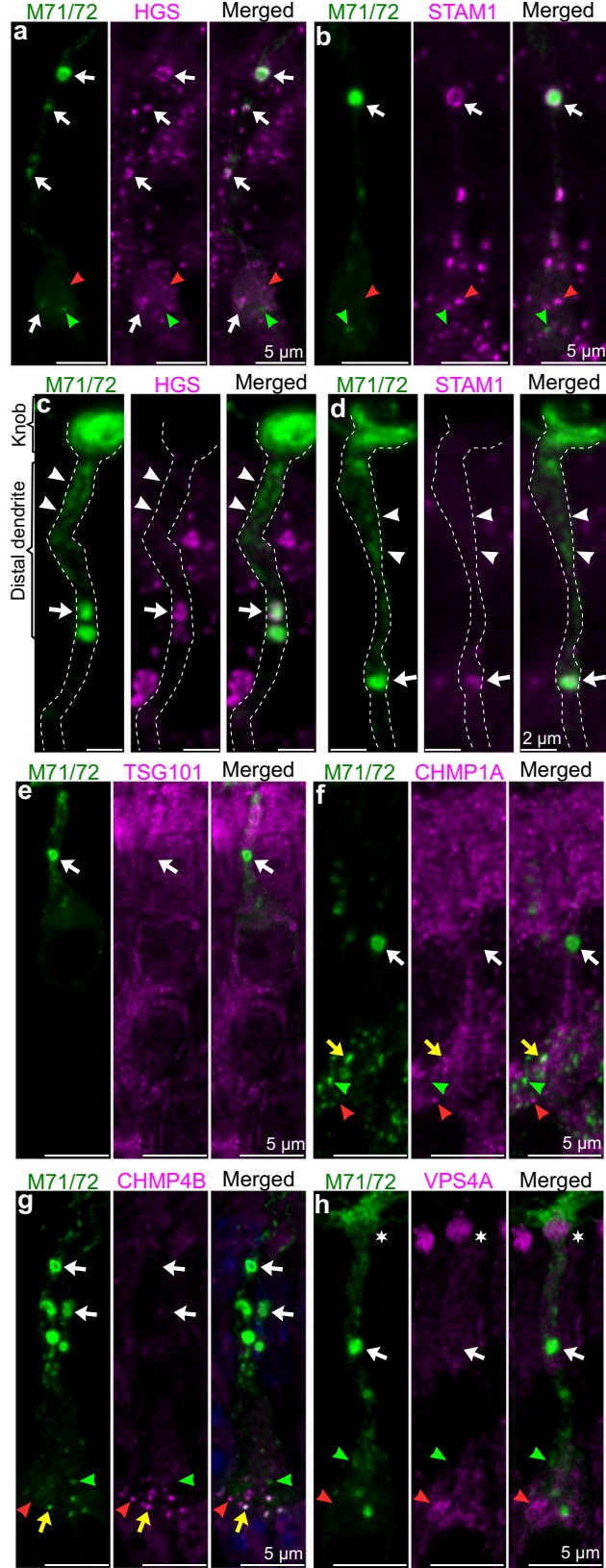

**Fig. 5 | The MVT is associated with ESCRT-0 but not downstream ESCRT complexes. a, b** Double immunohistochemistry showing M71/72+ puncta colocalized with the ESCRT-0 proteins HGS (5 mice, >60 OSNs) (**a**) and STAM1 (3 mice, >40 OSNs) (**b**) (arrows) in dendrites and the soma. The soma also contained M71/72+/HGS-, and M71/72+/STAM1- puncta (green arrowheads) as well as M71/72-/HGS+, and M71/72-/STAM1+ puncta (red arrowheads). **c, d** Enhanced exposure of M71/72/HGS (**c**) and M71/72/STAM1 (**d**) immunohistochemical signals showing MVTs with ESCRT-0 fluorescence (arrows) and MVTs lacking colocalization of M71/72/ESCRT-0 in the dendritic knob, plasma membrane and ILVs* in the distal dendritic segment (arrowheads). **e–h** Double immunohistochemistry showing that ESCRT-I (TSG101) (**e**), ESCRT-III (CHMP1A, CHMP4B) (**f, g**), and ESCRT-IV (VPS4) (**h**) did not colocalize with MVTs (arrows). However, M71/72/ESCRT-III colocalization was evident in the soma (yellow arrows in **f** and **g**). The soma also contained M71/72+/ESCRT-III- and M71/72+/ESCRT-IV- puncta (green arrowheads) as well as M71/72-/ESCRT-III+ and M71/72-/ESCRT-IV+ puncta (red arrowheads) (**f–h**). **e–h** Shown are representative images (3 mice, >15 OSNs).

MVTs. However, MVTs were present in a few OSNs that showed weak S100A5 signals (the gray sectors of the pie charts in Fig. 7e). These OSNs likely were in an early or late phase of odor-induced S100A5 accumulation at the time of tissue fixation. Notwithstanding, these results were compatible with the hypothesis that odor stimuli promoted the disintegration of the limiting membrane of MVTs in OSNs, resulting in the release of ILVs.

### Regulation of MVTs by forskolin-mediated activation of AC3

The naris occlusion experiment, in which S100A5 was used as an activity marker, revealed the long-term (days) effect of ambient odorants on MVTs. To analyze the possible short-term (sec) effect of olfactory transduction on MVT disintegration, we treated the OE with forskolin ex vivo. Similar to odorants, forskolin activates AC3 and the downstream olfactory transduction cascade[56]. Unlike odorants in inhaled air, which activate OSNs in a temporally uncontrolled manner, forskolin simultaneously activates all OSNs. To exclude OSNs that could be activated by both ambient odorants and forskolin, we focused on OSNs that were inactive (S100A5-) in ambient air. Accordingly, the changes in the numbers of the following categories of OSNs were quantified: (i) S100A5- OSNs containing MVTs (the red curve in Fig. 7f), (ii) S100A5- OSNs without MVTs (the yellow curve in Fig. 7f) and (iii) S100A5+ OSNs stimulated by ambient odorants (the black curve Fig. 7f). In agreement with the results of the naris occlusion experiment, forskolin treatment decreased the number of OSNs containing MVTs and conversely increased the number of OSNs without MVTs (Fig. 7f). Moreover, the result of this experiment involving "forced" activation of inactive OSNs showed that the disintegration of the limiting membrane occurred rapidly, i.e., within approximately 30 s. Note that the duration of the experiment (4 min) was too short to result in an increase in the number of S100A5+ OSNs, which would have hindered the analysis.

**Odorant-evoked MVT disintegration in freely behaving mice.** The naris closure and the forskolin experiments indicated that the MVT´s limiting membrane disintegrated in response to stimulation of olfactory signal transduction. To further address this idea we analyzed the number of RP2+ MVTs in odorant-exposed awake and freely behaving mice. The mouse OE is a mosaic of over thousand different OSN subpopulations, each expressing a defined OR that determines which odorants that subpopulation detects[57,58]. Acetophenone is an odorant agonist for M71/72 ORs[58,59]. Double immunohistochemical analysis for RP2 and M71/72 showed that exposing mice to acetophenone decreased the number of M71/72 OSNs with MVTs in the dendrite within minutes (Fig. 8). This result indicated that binding of an odorant to an OR evoked disintegration of the MVT's limiting membrane.

side was without a significant effect on MVT accumulation. The percentage of unstimulated OSNs that lacked MVTs and expressed RP2 in the distal dendritic plasma membrane was unaltered by naris occlusion (the yellow arrow and sectors of the pie charts in Fig. 7d´ and e, respectively). All OSNs that were intensely positive for S100A5 lacked

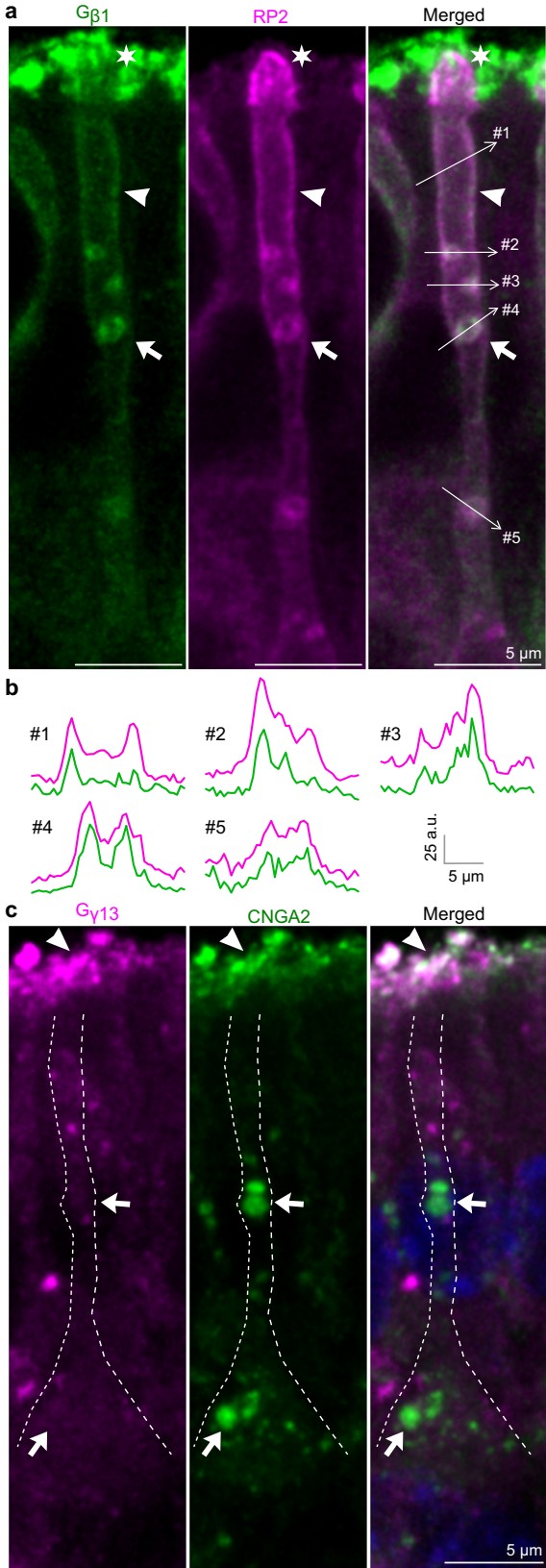

**Fig. 6 | Gβ1/RP2 colocalization and MVT-independent Gγ13 trafficking. a** Double immunohistochemistry showing Gβ1/RP2 colocalization in the limiting membrane (arrow), plasma membrane (arrowhead) and dendritic knob (asterisk). **b** Line scans of the plasma and limiting membranes showing a high degree of overlap between Gβ1 (green) and RP2 (magenta) immunofluorescence in areas of the dendrite that are indicated in (**a**) (#1–5). **c** Double CNGA2 (green)/Gγ13 (magenta) immunohistochemistry showing that Gγ13 and CNGA2 were colocalized in cilia (arrowhead) but not in the soma or dendrites (arrows). Representative images are shown (2 mice, >30 OSNs).

and contain ORs, CNGA2, AC3, Gβ1, Gαolf and TMEM16B which are proteins that directly mediate olfactory transduction within the ciliary compartment (the MVT pathway is schematically outlined in Supplementary Fig. 8). In accordance with this selectivity of cargo, we found that MVTs do not harbor TRPC2, which is a transduction channel present in the dendritic knob, not cilia[40]. Moreover, we previously showed that MVTs do not colocalize with the olfactory marker protein (OMP), which regulates transduction within the cytoplasm[38,40,60]. We find that ciliary "nontransduction" proteins, such as the intraflagellar transport components IFT20 and IFT88, as well as proteins enriched in the dendritic knob (e.g., ARL3 and VPS4A) are not present in MVTs. MVBs in general are associated with constitutive molecules that make up either the organelle structure or function in vesicle budding, ubiquitination, protein sorting or transport along microtubules[2]. Our previous study and the results herein show that MVTs are devoid of the coat proteins clathrin and caveolin as well as RABs and LAMP1/2, which regulate endocytic recycling and/or degradation[38]. We found that LAMP1/2 and lysosomes are located in the soma but not in dendrites or the dendritic knob. Thus, MVTs in the dendrite are segregated from late endocytic compartments in the cytoplasm of the soma. MVTs are negative for exosome proteins such as CD9, RAB11A/B, RAB35 and TSG101, GRASP55 and GRASP65, which are associated with the "compartment for unconventional protein secretion" (CUPS) and secretory/degradative autophagy pathways[38,61,62]. Instead, we found that MVTs are associated with synaptophysin and RP2, which are proteins that have not been previously associated with MVB-like organelles. These results show that MVTs are distinct from other types of MVBs, which are involved in conventional recycling, exocytic and endocytotic trafficking routes. Their large size, their association with synaptophysin, and the lack of both LAMP1 and luminal tubules also distinguish MVTs from previously described atypical KIFC2-associated somatodendritic MVB-like organelles as well as a type of late endosome that mediates ciliary targeting of peripherin 2 in photoreceptors[13,45].

MVTs are associated with ESCRT-0 (HGS and STAM1), which initiates the sequential assembly of MVBs by recognizing and binding ubiquitinated cargos, which are thereafter passed to downstream ESCRT complexes[3]. However, the maintenance of ESCRT-0 in the limiting membrane of assembled MVTs in dendrites does not fit with the conventional model of ESCRT-0 function. During the biogenesis of MVBs, ESCRT-0 disassociates from the limiting membrane prior to the phase in which ILVs are pinched off into the lumen of the MVB[3]. The HGS-STAM1 complex in the limiting membrane of the MVT might therefore persistently bind ubiquitinated proteins in the limiting membrane in a manner independent of its possible role in MVT biogenesis in the somatic cytoplasm.

MVTs can be in close contact with each other, and MVTs in the soma and the dendritic origin do not always contain all transduction proteins. This suggests that large MVTs may be generated by homotypic fusion. However, MVTs are not associated with Rab11, which promotes the formation of giant MVBs and exocytosis by regulating docking and membrane fusion[38,63]. Instead, we found that MVTs colocalize with synaptophysin and are closely associated with the RP2-binding protein NSF, which participates in the fusion of internal membranes by regulating SNARE proteins[52,53].

## Discussion

By combining in vivo CLEM with double and triple immunofluorescence and lectin staining and confocal microscopy, we show that most olfactory transduction proteins are transported and stored together in dendrites within large MVB-like dendritic organelles that we have named MVTs. MVTs disintegrate in response to odor stimuli

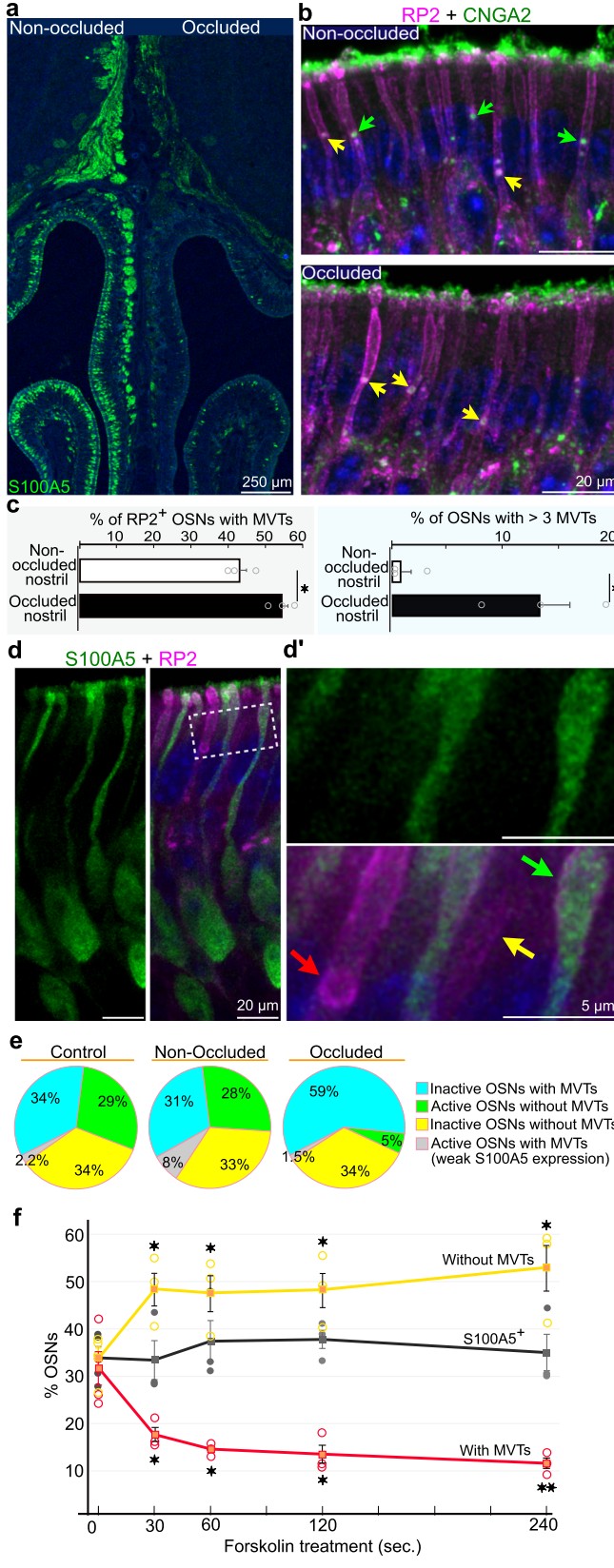

**Fig. 7 | Disintegration of the RP2⁺-limiting membrane is inhibited by naris occlusion and stimulated by the AC3 activator forskolin. a** S100A5 immunohistochemistry in OSN cell bodies and axons after unilateral naris occlusion is shown. The expression level of S100A5 (green) was reduced in OSNs located ipsilateral to the occluded naris (occluded) compared to OSNs located contralateral to the occluded naris (non-occluded). **b**, Double immunohistochemistry for RP2 (magenta) and CNGA2 (green) in OSNs in the OE of the non-occluded and occluded naris. MVTs (RP2⁺/CNGA2⁺ puncta) and ILVs* (RP2⁻/CNGA2⁺ puncta) are indicated by yellow and green arrows, respectively. **c**, The bar graphs show the percentage of OSNs with MVTs and the percentage of OSNs with > 3 MVTs in OE on the non-occluded and occluded side. Sample sizes were 509 and 539 OSNs for non-occluded and occluded nasal cavity, respectively. Two-sided unpaired Student's *t*-test gave $p = 0.020$ (left) and $p = 0.020$ (right) ($n = 3$ mice per condition). The values represent the mean ± SEM. **d** Double immunohistochemistry for RP2 (magenta) and S100A5 (green). **d'** Close-up of the area outlined in d. the red and yellow arrows indicate inactive (S100A5⁻) OSNs with and without an MVT, respectively. The green arrow indicates an active (S100A5⁺) OSN without an MVT. **e** Pie graphs showing the percentages of active/inactive OSNs with/without MVTs in OE of control (unoperated) mice and on the non-occluded and occluded side of unilateral naris occluded mice. Sample sizes were 630, 338 and 441 OSNs for control, non-occluded and occluded nasal cavities, respectively (3 mice per condition). **f** Graph showing the change in the number of inactive (S100A5⁻) OSNs with (red curve) and without (yellow curve) MVTs seconds after administration of forskolin to the OE ex vivo. The black curve indicates the number of S100A5⁺ OSNs. Sample sizes at 0, 30, 60, 120 and 240 s were; 1289 OSNs (4 mice), 957 OSNs (3 mice), 1095 OSNs (3 mice), 1164 OSNs (3 mice) and 1315 OSNs (3 mice), respectively. The values represent the mean ± SEM. Two-sided unpaired Student's *t*-test of data for 0 s compared to each of the time point indicated in the plot (not adjusted for multiple comparisons), gave the following p values from left to right: 0.026, 0.039, 0.029, 0.021 (curve in yellow) and 0.039, 0.017, 0.017, 0.009 (curve in red). *$p < 0.05$, **$p < 0.01$.

immunofluorescent puncta that, based on our CLEM results, lack a limiting membrane and thus correspond to clusters of released ILVs*. Our results further suggest that odorant-induced AC3 activation results in disintegration of the limiting membrane and release of ILVs* from the MVT into the cytoplasm of the apical dendritic segment. The RP2⁺-limiting membrane thus disintegrates at a location where the plasma membrane shows intense RP2 immunofluorescence. Such a reciprocal change in immunofluorescence between membranes is not evident for ESCRT-0 or synaptophysin. These results indicate that RP2 selectively translocates to the plasma membrane as a consequence of disintegration of the limiting membrane, although it cannot be excluded that the limiting membrane integrates with the plasma membrane at the same time. While the mechanism by which the transduction signal, such as increased cAMP/Ca²⁺ levels and/or depolarization, stimulates membrane disintegration and release of ILVs remains to be identified, our results show that this occurs on a timescale (<30 s) that is too rapid for alterations in gene expression to play a direct role. Unlike acetophenone and forskolin exposure, sensory deprivation by unilateral naris occlusion resulted in both an increased percentage of OSNs containing MVTs and an increased number of MVTs per OSNs. Together, these results indicate that MVTs function as reservoirs that in response to odorant stimuli release, transduction proteins destined for the ciliary compartment. This function of MVTs is also in line with the finding that RP2 is expressed in the limiting membrane. RP2 deficiency in mice results in mislocalization of cone opsin and reduced rhodopsin levels in the outer segment[64]. Studies in cultured cells have indicated that RP2 regulates vesicle transport between the Golgi and primary cilia. RP2 is a GAP for Arl3 that regulates the assembly and trafficking of membrane-associated protein complexes to primary cilia by stimulating the release of lipidated cargo proteins from their carriers, such as PDEδ[26,27,65,66]. Both RP2 and PDEδ are present in the dendritic knobs of OSNs, where they may interact with ARL3, which we found to be specifically localized to the ciliary base and proximal cilia.

Most MVTs are localized at the proximal start of a previously uncharacterized compartment in distal dendrites of OSNs. This compartment is characterized by RP2 expression in the plasma membrane that gradually increases toward the dendritic knob, where RP2 immunofluorescence is the most intense. The distal half of the dendrite normally lacks MVTs and instead contains smaller and weakly

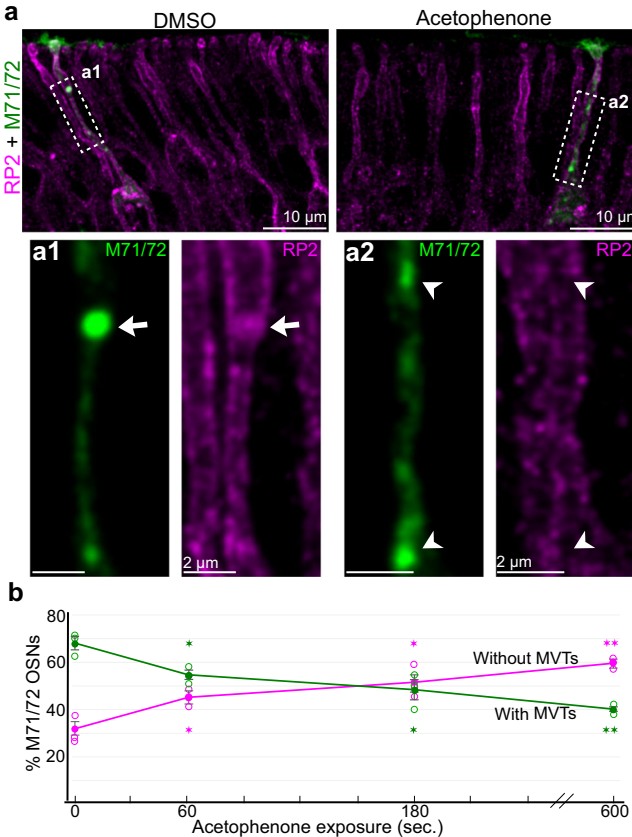

**Fig. 8 | Odorant-evoked disintegration of the RP2⁺-limiting membrane.**
**a** Double RP2 (magenta)/M71/72 (green) immunohistochemistry in OE obtained
after exposure of awake and freely behaving mice to DMSO or acetophenone/
DMSO for 5 min. **a1**–**a2** Close-up of areas outlined in (**a**). An intact MVT, i.e. RP2 and
M71/72 colocalization, is shown in (**a**), **a1** (arrow). Two M71/72⁺ puncta that do not
colocalize with RP2 are shown in (**a**), **a2** (arrowheads). **b** Line graph showing the
time course of percentage decrease and increase of M71/72 OSNs with and without
MVTs after acetophenone exposure. For each time point 0, 60, 180 and 600 s
sample sizes were 420, 393, 421 and 381, respectively, from 3 mice in each group.
The values represent mean ± SEM. Two-sided unpaired Student's *t*-test of data for
0 s compared to each of the time points indicated in the plot (not adjusted for
multiple comparisons), gave the following p values from left to right: 0.016, 0.018,
0.013 and 0.00068 for both curves. *$p < 0.05$, **$p < 0.01$.

RP2 and Gβ1 form an ARL3-regulated complex in photoreceptors,
which facilitates membrane association and trafficking of Gβ1[50]. We
found a high degree of overlap between RP2 and Gβ1, which suggests
that these proteins also form a complex in OSNs. Since RP2 is ubiqui-
tinated, it is possible that the binding of the ubiquitin moiety to ESCRT-
0 may allow loading of the RP2:Gβ1 complex on the limiting
membrane[67]. RP2 is also myristoylated at the N-terminal end, and this
modification is required for the translocation of RP2 from the intra-
cellular membrane to the plasma membrane[68]. This suggests that
exposure of the N-terminal following stimulus-induced membrane
disintegration may result in translocation of the RP2:Gβ1 complex to
the plasma membrane. Gαolf appears to be an ILV cargo, while Gγ13 is
associated with a dendritic transport pathway unrelated to MVTs.
These findings are consistent with a model in which one function of
MVTs is to physically and temporally segregate the different Golf
subunits from each other prior to Gαolf:Gβ1:Gγ13 heterotrimerization
in the dendritic knob, which is coordinated by proteins such as ARL3,
PDEδ, Ric-8B and CEP290[18,32,50].

In summary, our study identifies an odorant-regulated vesicular
pathway in the dendrites of OSNs in which MVTs transports and stores
ciliary olfactory transduction proteins and harbors constitutive

proteins such as ESCRT-0, RP2, and synaptophysin. The results suggest
that MVTs play an important regulatory role in the compartmentali-
zation and sorting of GPCR signaling components in dendrites. These
findings provide evidence for the existence of a previously unknown
regulatory protein trafficking mechanism in neurons, laying the
foundation for future studies on stimulus-regulated dendritic trans-
port, storage and assembly of GPCR signaling components between
the Golgi exit and postsynaptic membranes.

## Methods
### Animals
All animal experiments were approved by the Local Ethics Committee
for Animal Research at the Court of Appeal for the upper northern area
of Norrland (Umeå, Sweden). The mice were kept in IVCs on a 12 h
light/dark cycle at the Umeå Center for Comparative Biology, Umeå.
C57Bl/6 J mice of both sexes were analyzed. The mice were sacrificed
by cervical dislocation followed by immediate decapitation.

### Naris occlusion
Unilateral naris occlusion in mice was achieved by cauterization. At
postnatal day five, five pups were anesthetized by subcutaneous
administration of a cocktail containing 0.25 mg/kg medetomidine,
2.5 mg/kg midazolam, and 0.025 mg/kg fentanyl. The right nostril of
each deeply anesthetized pup was cauterized with a mini electro-
surgery unit controlled bipolar forceps (Martin, Germany). Immedi-
ately after cauterization of the nostril, the mice were given an antidote
(cocktail of 1.25 mg/kg atipamezole and 0.6 mg/kg naloxone) sub-
cutaneously. The pups were transferred to their mother's cage, and
formation of a scar that resulted in complete unilateral naris closure
was confirmed by applying a drop of water to the operated nostril and
by postmortem examination. Naris closure was maintained for 7 days.

### Forskolin and vismodegib treatment
A solution of 50 μm forskolin (CAS: 66575-29-9, catalog: 1099, Tocris,
Biotechne) was prepared in fresh Ringer's solution (140 mM NaCl,
5 mM KCl, 1 mM CaCl2, 1 mM MgCl2, 10 mM HEPES, 10 mM glucose,
1 mM sodium pyruvate; pH 7.4). Mice were decapitated and the head
cut in half along the sagittal, interfrontal and internasal sutures. Then,
the nostril septum was removed. One half of the head was incubated in
forskolin solution or Ringer's solution at room temperature (25 °C) for
30, 60, 120 and 240 s (i.e. the time points given in Fig. 7f). At the end of
each time point the tissue was washed in PBS and fixed in 4% paraf-
ormaldehyde in PBS (PFA; VWR Chemicals, 28794.295; w/v in PBS).
Time from start of PBS wash to fixation was ~10 s. Five 40 mg/kg i.p.
doses of vismodegib (CAS: 879085-55-9, catalog: GDC 0449, Sell-
eckchem) was administered with 12 h intervals and en face prepara-
tions were generated to analyze immunofluorescence in cilia[38].

### Odorant exposure
Two weeks old mice were exposed to 50 μl 50% acetophenone (CAS:
98-86-2, catalog: 42163, Sigma-Aldrich) in DMSO (CAS: 67-68-5, cata-
log: D4540, Sigma-Aldrich) or DMSO essentially as described[69]. In
short, the mouse was placed in a 10x10x8 cm odorless plastic box fitted
with a lid and a clean 1 x 1 x 0.3 cm 3MM Whatman paper. The mouse
was habituated for 5 min, acetophenone or DMSO was then placed on
the Whatman paper and the lid was closed. Mice were sacrificed after
60, 180 or 600 s exposure (time points given in Fig. 8b) and the nasal
tissue was fixed 75–90 s. later in 4% PFA.

### Preparation of nasal tissue for confocal microscopy
Nasal tissue was dissected, fixed for 4 h in 4% PFA, incubated in 30%
sucrose (w/v in PBS) overnight and frozen in OCT (Histolab, Gothen-
burg, Sweden). The OCT blocks were sectioned using an HM 550
Cryostat Microtome (Microm) at a thickness of 14 μm, air-dried, incu-
bated with citrate buffer (10 mM, pH 6.0) for 5 min at 100 °C, washed in

PBS and incubated in blocking solution (2% FCS + 0.2% Triton X-100 in PBS). The sections were washed in PBS, incubated overnight with primary antibodies in blocking solution, washed and incubated with secondary antibodies diluted in T-PBS (0.1% Tween-20 in PBS). The sections were then stained with Hoechst (0.1 μg/ml) and mounted with fluorescence mounting media (Dako, CA, USA). The details of antibodies used in this study are in supplementary table 1. The lectins biotinylated DBA (1:300 Sigma, L6533) and WFA (1:100, Sigma, L1766) were visualized with Cy™3 Streptavidin (Jackson ImmunoResearch Europe, 016-160-084). For immunohistochemical analyses results from biological replicates (numbers in figure legends) gave similar results. Imaging was performed with a Leica TCS SP8 confocal system equipped with a Leica DMi8 microscope using an HC PL APO CS2 63x/ 1.40 N.A. objective. LAS X software was used to acquire confocal images with a voxel size of 0.60 μm and a pixel size of either 0.09 × 0.09 or 0.12 × 0.12 μm². Confocal images with a z-axis thickness of ~8 μm were acquired using LAS X software.

## Tissue preparation for CLEM

Tissue was dissected and incubated for 30 min in fixative (2% PFA, Fisher Scientific PA0095; 0.2% glutaraldehyde, TAAB Laboratories Equipment G011/2) in 0.1 M phosphate buffer (PB) pH 7.4. The OE from turbinates was fixed by incubation for 10 min at 20 °C in a vacuum (20 Hg) using the PELCO BioWave® Pro+ Microwave Processing System. The tissue was then washed three times for 5 min in PBS and one time for 5 min in 1% glycine/PBS (Merck, cat. No. 104204.0100), transferred to a 1 mm thin sheet of 12% gelatin, and incubated for 30 min at 37 °C. The gelatin-embedded tissue was cut into 1 mm³ blocks, cryoprotected overnight in 2.3 M sucrose (VWR Chemicals, 27480.294) in 0.1 M PB and frozen in liquid nitrogen. Ultrathin (75 nm) sections were generated at −120 °C with a Leica UC7 ultramicrotome in a cryochamber. The sections were transferred to a drop of a 1:1 mixture of 2% methyl cellulose (Sigma, M-7140), 2.3 M sucrose and 0.1 M PB and mounted on TEM grids with a carbon-coated Formvar film (TAAB Laboratories Equipment, F005). The grids were incubated in PBS with the sections facing the PBS at 37 °C for 30 min, rinsed in PBS, incubated in blocking solution (1% FCS, 0.1% gelatin (w/v) in PBS) for 10 min and incubated overnight in blocking solution containing primary antibody or lectin. The grids were washed six times in blocking solution, incubated at room temperature for 1 h in blocking solution containing secondary antibody or streptavidin, counterstained for 5 min in Hoechst/PBS (0.1 μg/ml), washed three times with blocking solution, washed four times in Milli-Q® water and mounted on microscope slides in 50 μl of 50% glycerol (Sigma Aldrich, cas no. 56-81-5). Scanning was performed with a Leica TCS SP8 confocal system equipped with a Leica DMi8 microscope and LAS X software. The overview and ROIs were scanned using an HC PL APO CS2 20x/0.75 IMM objective and HC PL APO CS2 63x/1.40 N.A objective, respectively. The scanned grids were washed seven times with Milli-Q water, contrasted for 10 min, transferred and embedded in 1.8% methyl cellulose/0.4% uranyl acetate (MC/UA, pH 4, Polysciences, Inc., cat. No. 21447-25) on cold plates in the dark.

## CLEM

TEM was performed with a Talos 120 C transmission electron microscope (FEI Company) operating at 120 kV. Maps 3.3 application software (FEI Company) was used to correlate confocal microscopy images with electron micrographs, which were acquired with a Ceta 16 M CCD camera (FEI Company), Maps 3.3 application software, Velox Software (FEI Company) and TIA with a 1 s exposure and a resolution of 4000 × 4000 pixels. CLEM alignments and analyses from different biological replicates (details are in figure legends) gave similar results. Reference points for the alignment of stitched low magnification (1600X or 2600X) TEM images and confocal images, were electron dense nuclear contrast (heterochromatin) and intensely stained

nuclear Hoechst fluorescence in TEM and confocal images, respectively (supplementary Fig. 2).

## Image analysis

Confocal and TEM images were aligned using the "Big Wrap" tool within the ImageJ image processing package Fiji[70]. X-Y-Z images and 3D rotation images were obtained and colocalization confirmation was achieved with 3D Visualization-Assisted Analysis Suite (Vaa3D) software version 3.601[71] or LAS X software (Leica Microsystems). The localization of immunofluorescent puncta was determined using the "Plot profile" in Fiji[70]. The sizes of the MVTs and ILVs and the number of ILVs per MVT section were determined from CLEM images of 97 OSNs acquired from seven mice using the 'Measure' and 'Cell counter' tools, respectively, in Fiji. The fluorescence area (size), fluorescence intensity and fluorescence localization were determined from images of 166 OSNs from 5 mice using the same confocal scanning settings. Z-stack images were collapsed using the maximum intensity projection method 'z-project' in Fiji[70]. For analysis of MVBs in occluded and non-occluded nostrils, sections from three mice were used. Four OE areas (108 × 108 μm) of occluded and non-occluded nasal cavities were analyzed. The same regions of the OE from 3 mice were scanned on the occluded and non-occluded sides. In the experiments shown in Fig. 7c, the numbers of analyzed OSNs in the occluded and non-occluded nasal cavities of 3 mice (>150 OSNs/mouse) were 509 and 539, respectively. In the experiment shown in Fig. 7e, the numbers of analyzed OSNs in the control (unoperated), occluded and non-occluded nasal cavities of 3 mice (>120 OSNs/mouse) were in total 630, 441, and 388, respectively. In the experiment shown in Fig. 7f, the numbers of analyzed OSNs from 3–4 mice at time points 0, 30, 60, 120 and 240 s were in total 1289, 957, 1095, 1164 and 1315, respectively. Quantification of M71/72 OSNs with or without MVTs following acetophenone exposure was done on 3 mice, six coronal OE sections per mice, for each time point. Sections were separated by 140 μm. In the experiment shown in Fig. 8b, the number of analyzed OSNs at time points 0, 60, 180 and 600 s were in total 420, 393, 421 and 381, respectively. Images were assembled using Inkscape version 1.1.2.

## Statistical analysis

The data are expressed as the mean ± SEM. Student's unpaired $t$-test was used. A $p$ value >0.05 indicated nonsignificance (n.s.). Microsoft Office Excel 2013 was used for statistical analysis. R version 4.0.5 was used to generate boxplots. Three or more mice from each experimental group were analyzed. The sample size was >100 for comparisons of OSN number and MVB size. No statistical method was used to determine the sample size. The sample sizes were in accordance to standard in field and the 3 R principle. No randomization method was used to collect the data. None of the data points were excluded from the study. For the measurements shown in Figs. 7 and 8, the experimenter was blind to experimental group allocation during data collection and analysis.

## Reporting summary

Further information on research design is available in the Nature Portfolio Reporting Summary linked to this article.

# Data availability

Data that support the findings of this study is present in the paper, supplementary information, Source Data and the Zenodo database (10.5281/zenodo.7194768). Source data are provided with this paper.

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

## Acknowledgements

The authors acknowledge the facilities and technical assistance of the Umeå Core Facility Electron Microscopy (UCEM) at the Chemical Biological Centre (KBC), Umeå University. The authors also thanks Dr. Gilad Barnea (Department of Neuroscience, Brown University, Providence, USA) for the anti-M71/72 antibody and Dr. Claus W. Heizmann (Department of Pediatrics, University of Zürich, Zürich, Switzerland) for the anti-S100A5 antibody. This work was supported by The Kempe foundation (JCK-1619 to S.B.), the medical faculty at Umeå University (FS 2.1.6-1119-19 B1 to S.B.) and D.K.M. was supported by a postdoc grant from the Swedish Society for medical research (SSMF-322018727).

## Author contributions

D.K.M. planned and performed experiments, analyzed data and wrote the manuscript. A.B. and S.B. planned the experiments, supervised the study and wrote the manuscript.

## Funding

## Competing interests

The authors declare no competing interests.
