## [Peer Review File · Nature Communications]

A multivesicular body-like organelle mediates stimulus-regulated trafficking of olfactory ciliary transduction proteinsReviewer #1 (Remarks to the Author):

The manuscript by Maurya and colleagues addresses the transport of olfactory transduction proteins in the dendrites of olfactory sensory neurons. They are using a combination of immunofluorescent and electron microscope to show the nature of organelle that shuttles proteins along and how its fate is affected by increased cAMP levels. Their results are novel, interesting, timely and mostly well supported by their experiments.

Two major points

- 1 The authors use forskolin as a substitute for odorant stimulation. But this experiment might be ambiguous. Forskolin does activate the olfactory transduction cascade at the level of the ciliary adenylyl cyclase, that cause depolarization that might trigger other (non-cAMP) events that trigger the disintegration of the limiting membrane. Or it could be that forskolin activates adenylyl cyclases in the dendrite to cause an increase in cAMP there (it is probably unlikely that cAMP generated in the cilia diffuses half the way down the dendrite). Along similar lines, in the figure legend of Suppl Fig. 7 the authors talk about "cAMP-dependent disintegration", suggesting that the authors think that is actually a cAMP-mediated mechanism? A potentially interesting and elegant way to address this issue would be, as the authors have an antibody for the M71/72 odorant receptors, to stimulate with M71 or M72 agonists and study MVTs in M71/72 olfactory sensory neurons.
- 2 As naris occlusion changes olfactory physiology in both the open and the occluded size the authors should include data from mice that did not have their naris occluded. This way it will actually be clear what is up or downregulated compared to the normal state.

Minor comments:

- 1 Although the authors describe the numbers of their experiments in the methods, it might be useful to also include them in e.g. the figure legends to understand what the data points mean. E.g. in Fig. 7C there seem to be three data points. Are those from three mice?
- 2 Page 10, line 12. Panels Fig. 4g-h do not seem to get mentioned?
- 3 Page 14, line 18: "the green graph Fig. 7f". It looks black to this reviewer?
- 4 Page 18, line 20: "replenish these proteins in cilia". As the authors are actually not showing this, they might want to moderate the statement.
- 5 Page 21, line 21: how long between Forskolin stimulation and fixation?
- 6 Suppl. Fig 1: confocal is misspelled.
- 7 Page 50, line 9: know -> knob.

Reviewer #2 (Remarks to the Author):

In this paper by Maurya et al. immunofluorescence and CLEM are used to investigate the trafficking and sub-cellular localization of proteins involved in sensory transduction in olfactory neurons. Specifically, the authors show that many key proteins including odorant receptors and cyclic nucleotide-gated ion channels are localized in large dendritically positioned puncta. This was first shown by the same authors previously (Maurya et al. 2017 PNAS). Now, using CLEM, the authors show that these transport puncta are large multivesicular-like vesicles. Specifically, they are ~600nm sized vesicle filled with smaller ~60 nm diameter intraluminal vesicles. The authors demonstrate that these MVB-like vesicles tend to localize mid-way up the dendrite and associate with specific ESCRT proteins. Lastly, these vesicles burst when localized towards the end of the dendrite, presumably to deliver important signaling proteins and channels to the plasma membrane to modulate the signaling behavior of the neuron. In general, I find this paper interesting. The authors clearly show that key proteins are sequestered and transported in the dendrites by these multivesicular body-like organelles. The finding that these organelles lose their limiting membrane and likely deliver smaller protein-containing vesicles to the dendritic knob is interesting. The immunofluorescence and CLEM is straightforward and the findings clear and well described. I have several suggestions and comments that might improve the manuscript.

1. In general the quality of the EM in the CLEM images is difficult to evaluate. I would recommend that the authors present more extensive "overview" images of as much of the cell as possible without the fluorescence overlay. I cannot, for example, evaluate the quality of the EM or

correlations from Figure 1g and g' with the currently displayed zooms and contrasts.

2. The paper presents the quantitative numbers on how many cells/animals/neurons were analyzed in the experiments in the methods section. I would suggest that the authors present how many animals/cells/neurons were analyzed and how often these structures were observed in the main text or figure legends.

3. I don't think this is correct

"Data availability. All data obtained during this study are included in the manuscript"

All of the imaging data is not presented or available in the paper. Please change.

4. Red/green color coding is difficult for color blind readers to evaluate (~10% of men). I would recommend changing to a different color blind-friendly color scheme.

5. The findings from the occluded/non-occluded experiments are particularly nice. I appreciate these internally-controlled experiments.

6. Can the authors please expand on how the CLEM correlation was done and what fiducials or markers were used to ensure that the correlation was done correctly and within a reasonable amount of alignment error.

Reviewer #3 (Remarks to the Author):

This manuscript describes a new type of multivesicular body-like organelles in the dendrite of olfactory receptor neurons (ORNs). Using immuno/chemical fluorescence confocal microscopy and CLEM, the authors show that these ORN dendritic MVBs contain most olfactory transduction proteins. These MVBs contain ESCRT-0 proteins but not the downstream ESCRT complex proteins. These MVBs are associated with RP2 and synaptophysin, which are not found in other types of MVBs, but do not contain various proteins that label other types of MVBs and other membrane organelles. The authors further show that the activation of the transduction pathway can regulate the quantity and the disintegration of these MVBs. The study thus suggests a previously unknown cellular mechanism for dendritic trafficking of olfactory transduction proteins.

The finding in this study is novel, although the biological significance of these MVBs to olfaction (i.e., whether and how much this MVB-mediated trafficking of olfactory transduction proteins contributes to the olfactory response and/or the changes of the response) remains unknown. The overall quality of the imaging and analysis is high. It is exciting to see that CLEM is now used to study ORN cell biology and make discoveries.

The following points are for the authors to consider,

1. Immunostaining is the primary technique in this study, and the authors have used a few tens of antibodies. The specificities of these antibodies should be addressed wherever possible (e.g., citations, manufacturers' confirmation in knockout mice, or authors' own test).

2. The authors tried to cover all olfactory transduction proteins from the receptor to the CNG channel. However, it is well-established that olfactory transduction involves a calcium-activated chloride channel, TMEM16B. The chloride current constitutes a major portion of the transduction current. Although the behavioral role of this channel remains unsettled, its role in signal amplification in the transduction is well-established. Why do the authors completely ignore this transduction component?

3. Only one anti-OR antibody (i.e., anti-M71/M72) was used in this study, so make this point clear in the text and delete anti-Mor28 from the acknowledgment. Accordingly, all the statements regarding OR should be specific. For example, the statement "Each OSN in the proximal half of dendrites contains a few MVTs" (Page 9, line 3) should be "Each M71- or M72-expressing OSN ...",

because Fig 7 shows that a substantial portion of ORNs, even in unstimulated conditions, are absent of MVTs, while the authors state "the majority of M71- or M72-positive OSNs (> 90%) contained one or up to three distinctive MVTs..." Also, does M71/M72 MVB staining overlaps with any of the DBA and WFA staining?

4. Naris occlusion was done in pups, not in adult animals. The authors should make this clear when describing the results.

5. What is the rationale for testing PDE4D, which is sustainably stained in the soma and dendrite in Suppl Fig. 4? Is it known that PDE4D is broadly expressed in ORNs? Is the antibody specificity confirmed?

6. An arrowhead in Fig 1b merged is not aligned.

Point-by-point response to Reviewer comments:**REVIEWER #1**

We appreciate Reviewer 1's positive comments and thank him/her for helpful and constructive comments that have significantly improved our manuscript.

Major comments

1. We agree that the suggested experiment using a M71/72 agonist is interesting and elegant. Accordingly, we have now included the experiment (Fig. 8). The result is that minutes of the M71/72 agonist (acetophenone) exposure of awake mice resulted in disintegration of the MVT's limiting membrane.

We also agree to that our results do not show which event downstream of adenylyl cyclase that actually triggers the disintegration of the membrane. We have changed in the text, which now states: "*While the mechanism by which transduction signal, such as increased cAMP/Ca²⁺ levels and/or depolarization, stimulates membrane disintegration and release of ILVs remains to be identified, our results show.....*"(page 18, line 14). We have also replaced "*cAMP-dependent disintegration*" by "*odorant-induced disintegration*" in the legend to Suppl. Fig 8 and in the Suppl. Fig. 8, "*cAMP*" is changed to "*signal transduction events*".

2. To distinguish whether a change in MVTs is caused by increased or absent odor stimulation (or both), we include result from unoperated control mice. The result show that the ratio of neurons with or without MVTs is similar in unoperated control mice and on the non-occluded side of operated mice. The results thus indicate that lack of odor stimuli causes accumulation of MVTs (Fig. 7e, page 14, line 1).

Minor comments:

1. The numbers of the experiments is now included in the figure legends and the data points for individual mice are now added (including in Fig. 7c).

2. We now refer to Fig. 4g-h on page 10 line 1. However, the result that transduction proteins in the soma both did and did not colocalize with RP2 (which is shown in Fig. 4g-h) is discussed in conjunction with ESCRT proteins and potential biogenesis of MVTs in the soma on page 11 lines 7-14.

3. Yes, the "green graph" in Fig. 7f was black (now changed on page 14, line 22).

4. We have deleted "*replenish these proteins in cilia*" and changed the sentence to "*Together, these results indicate that MVTs function as reservoirs that in response to odorant stimuli, release transduction proteins destined for the ciliary compartment*". Page 18, line 19.

5. Time from start of PBS wash to fixation was ~ 10 sec. This is now stated on page 21 line 22.

6. "Confocal" is now correctly spelled (Suppl. Fig. 2, now Suppl. Fig. 3).

7. "know" is changed to "knob".

REVIEWER #2

We thank Reviewer 2 for the positive comments about our manuscript and suggestions of how to improve the study. We have addressed the points raised as follows:

- 1.** To facilitate the evaluation of the EM quality in the CLEM images we have added EM images of cells at lower magnifications of without fluorescence overlay. The images show the neuron shown in Fig. 1g' an additional neuron (Suppl. Fig. 2).
- 2.** The numbers of animals/cells analyzed are now included in figure legends. How often MVTs were observed in OE under different conditions is best appreciated in the pie charts in Fig. 7e, which now also includes results from unoperated control mice.
- 3.** It is correct that all of the imaging data is not presented or available in the paper. On page 27 line 6 we now state; *"The data that support the findings of this study is present in the paper, supplementary information, Source Data or is available from the corresponding author upon reasonable request"*.
- 4.** We have changed the red/green color coding to magenta/green color coding.
- 5.** For the occluded/non-occluded experiment we have added data from unoperated control mice (see comment 2, reviewer #1).
- 6.** The CLEM correlation was done using nuclear immunofluorescence and the electron dense nuclear contrast as reference points. This is now illustrated in Suppl. Fig. 2 and is explained in the legend to Suppl. Fig. 2, page 46 as well as on page 25 line 15. Our experience is that this CLEM alignment strategy worked at magnifications up to 28000X.

REVIEWER #3

We appreciate Reviewer 3's positive comments and thank him/her for helpful and constructive criticism. Our responses are:

1. We have included a table with information on antibody specificities (Suppl. Table 1).
2. Yes, we agree that it is interesting to determine whether the Ca²⁺-activated chloride channel TMEM16B also is present in the MVTs. We have now included a double immunohistochemical analysis to show that this indeed is the case (Fig. 1e, page 6, line 6).
3. We agree that all the statements regarding ORs should be specific. Accordingly, we have deleted anti-Mor28 from the acknowledgment and changed from "*Each OSN in the proximal half of dendrites contains a few MVTs*" (Page 9, line 4) to "Each M71/72-expressing OSN ...". In the text and Figs we have also changed ORs to M71/72 when applicable. Moreover, we added that DBA and WFA staining are distinct from M71/72-expressing OSNs (page 8, line 12).
4. That "*12 day old mice that had been subjected to unilateral naris occlusion from postnatal day 5*" is now included in the result section on page 13 line 14-15.
5. We tested PDE4D because it regulates rapid termination of olfactory transduction¹. Another reason is that PDE4D interacts with PDE4DIP (myomegalin). PDE4DIP is an anchor for different proteins in the cAMP pathway and analyzing these two proteins was interesting in light of the forskolin-induced disintegration of the MVT's limiting membrane. PDE4DIP (which was expressed in a similar pattern as PD4E), has also been shown to regulate microtubule-dependent cargo transport in the inner segment towards the ciliary base of photoreceptors². A third reason was that the PDE4DIP:PDE4 complex regulates hedgehog signaling, which also is involved in odorant receptor transport^{3,4}. The anti-PDE4D antibody we used is validated by knockdown experiments in two published studies^{5,6}. Cygnar et al. have analyzed the distribution of PDE4D in olfactory sensory neurons¹. Although they used another antibody, their result is similar to ours, showing staining in soma and dendrite but not in cilia.
6. The arrowhead in Fig 1b is now aligned.

References

1. Cygnar, K. D. & Zhao, H. Phosphodiesterase 1C is dispensable for rapid response termination of olfactory sensory neurons. *Nat Neurosci* **12**, 454-462, doi:10.1038/nn.2289 (2009).
2. Overlack, N. et al. Direct interaction of the Usher syndrome 1G protein SANS and myomegalin in the retina. *Biochim Biophys Acta* **1813**, 1883-1892, doi:10.1016/j.bbamcr.2011.05.015 (2011).
3. Peng, H. et al. Myomegalin regulates Hedgehog pathway by controlling PDE4D at the centrosome. *Mol Biol Cell* **32**, 1807-1817, doi:10.1091/mbc.E21-02-0064 (2021).
4. Maurya, D. K., Bohm, S. & Alenius, M. Hedgehog signaling regulates ciliary localization of mouse odorant receptors. *Proc Natl Acad Sci U S A* **114**, E9386-E9394, doi:10.1073/pnas.1708321114 (2017).
5. Ren, H. et al. PDE4D binds and interacts with YAP to cooperatively promote HCC progression. *Cancer Lett* **541**, 215749, doi:10.1016/j.canlet.2022.215749 (2022).
6. Lu, M. Y. et al. Upregulation of miR-219a-5p Decreases Cerebral Ischemia/Reperfusion Injury In Vitro by Targeting Pde4d. *J Stroke Cerebrovasc Dis* **29**, 104801, doi:10.1016/j.jstrokecerebrovasdis.2020.104801 (2020).

Reviewer #1 (Remarks to the Author):

Maurya et al revised their manuscript nicely and responded well to concerns raised. The newly added data further strengthens the manuscript.

One point of confusion is that the authors state "We tested PDE4D because it regulates rapid termination of olfactory transduction" and cite Cygnar and Zhao in support. But in this paper PDE4D is not addressed, instead PDE4A and PDE1C, with both playing only a small role in rapid response termination. So why did the authors stain for PDE4D?

Reviewer #2 (Remarks to the Author):

The authors have addressed my previous comments in their revisions. I have two small issues that the authors could address.

1. The new "MVT" term seems unnecessary. If this is truly a variant of a "MVB" I would recommend the authors use the more common term and not coin a totally new term just for this structure in these cells.

2. It is unclear if the authors counted the total number of ILVs in a 3D MVB or if this is just the number in one thin section image (Figure 2b). If this number is the number of ILVs in a single thin section, please indicate this and then estimate the total number that would be expected from the entire 3D space of a MVB in the text.

Reviewer #3 (Remarks to the Author):

I appreciate the authors' effort in addressing my questions. I do not have additional questions regarding the revised manuscript, except for the expression of PDE4D in olfactory receptor neurons. The authors may have confused between PDE4A and PDE4D, which are different PDE proteins coded by different genes. Cygnar et al. studied PDE4A, rather than PDE4D. Several published works show the expression of the PDE4A in olfactory receptor neurons but none for the PDE4D. Also, that "PDE4DIP:PDE4 complex regulates hedgehog signaling" and that (hedgehog signaling) "is involved in odorant receptor transport" are independent facts, and they do not necessarily mean that PDE4D is involved in olfactory receptor neurons. The authors should either clarify this issue by double confirming the expression of PDE4D, maybe as well as PDE4DIP, using a different approach and not merely rely on the immunostaining with the particular antibody used, or remove the data for PDE4D, and maybe PDE4DIP as well, from the supplementary figure. This is not a critical piece of data for the manuscript, but if included, it has to be unambiguous.

Point-by-point response to the reviewers' comments

Reviewer #1 (Remarks to the Author):

Maurya et al revised their manuscript nicely and responded well to concerns raised. The newly added data further strengthens the manuscript.

One point of confusion is that the authors state "We tested PDE4D because it regulates rapid termination of olfactory transduction" and cite Cygnar and Zhao in support. But in this paper PDE4D is not addressed, instead PDE4A and PDE1C, with both playing only a small role in rapid response termination. So why did the authors stain for PDE4D?

Authors reply to reviewer #1

As suggested by reviewer #3, we have removed PDE4D in the revised version of the manuscript.

The reason why we analyzed for PDE4D was that it could have been an MVT-associated transduction protein because it, like other phosphodiesterases, may degrade odor-induced cAMP. In the response to reviewer #3 we also mentioned two additional hypotheses which led us to analyze for PDE4D. We now realize that we answered reviewer #3 incorrectly by referring to a publication about odor response termination and PDE4A and PDE1C but not PDE4D.

Reviewer #2 (Remarks to the Author):

The authors have addressed my previous comments in their revisions. I have two small issues that the authors could address.

1. The new "MVT" term seems unnecessary. If this is truly a variant of a "MVB" I would recommend the authors use the more common term and not coin a totally new term just for this structure in these cells.

2. It is unclear if the authors counted the total number of ILVs in a 3D MVB or if this is just the number in one thin section image (Figure 2b). If this number is the number of ILVs in a single thin section, please indicate this and then estimate the total number that would be expected from the entire 3D space of a MVB in the text.

Authors reply to reviewer #2

1. The MVT morphology is similar to MVBs, but the function is distinct from that of MVBs. It carries the select cargo of transduction proteins. The naming is in analogy multivesicular body-like organelles such as MHC II-enriched compartments (MIICs) and first stage melanosomes, which like the MVT are MVB variants that are not part of generic recycling/degradative/exosome pathways. Moreover, the MVT differs from previously characterized MVBs and MVB-like organelles in that it is regulated by stimulus activity and constitutively harbors RP2, synaptophysin and ESCRT-0. However, we leave it to the discretion of the editor whether we should change the name of the MVT.

2 We thank reviewer #2 for helpful comments. It is now stated in Fig. 2b that ILV numbers are per section. In Fig. 2 legend the number of ILVs per area and ILV numbers for 4 representative ILVs within the interquartile range, are stated.

Reviewer #3 (Remarks to the Author):

I appreciate the authors' effort in addressing my questions. I do not have additional questions regarding

the revised manuscript, except for the expression of PDE4D in olfactory receptor neurons. The authors may have confused between PDE4A and PDE4D, which are different PDE proteins coded by different genes. Cygnar et al. studied PDE4A, rather than PDE4D. Several published works show the expression of the PDE4A in olfactory receptor neurons but none for the PDE4D. Also, that "PDE4DIP:PDE4 complex regulates hedgehog signaling" and that (hedgehog signaling) "is involved in odorant receptor transport" are independent facts, and they do not necessarily mean that PDE4D is involved in olfactory receptor neurons. The authors should either clarify this issue by double confirming the expression of PDE4D, maybe as well as PDE4DIP, using a different approach and not merely rely on the immunostaining with the particular antibody used, or remove the data for PDE4D, and maybe PDE4DIP as well, from the supplementary figure. This is not a critical piece of data for the manuscript, but if included, it has to be unambiguous.

Authors reply to reviewer #3

Indeed we mixed up PDE4A and PDE4D and we agree to that a possible involvement of the PDE4DIP:PDE4 complex in hedgehog-regulated transport are independent facts. As we do not mention or discuss this preliminary negative data in the manuscript we have done as suggested and removed the PDE4D and PDE4DIP from supplementary data.